# HPLC-MS/MS Oxylipin Analysis of Plasma from Amyotrophic Lateral Sclerosis Patients

**DOI:** 10.3390/biomedicines10030674

**Published:** 2022-03-15

**Authors:** Mauricio Mastrogiovanni, Andrés Trostchansky, Hugo Naya, Raúl Dominguez, Carla Marco, Mònica Povedano, Rubèn López-Vales, Homero Rubbo

**Affiliations:** 1Departamento de Bioquímica, Facultad de Medicina and Centro de Investigaciones Biomédicas, Universidad de la República, Montevideo 11800, Uruguay; maurimastro@fmed.edu.uy (M.M.); trocha@fmed.edu.uy (A.T.); 2Unidad de Bioinformática, Institut Pasteur-Montevideo, Montevideo 11400, Uruguay; naya@pasteur.edu.uy; 3Departamento de Producción Animal y Pasturas, Facultad de Agronomía, Universidad de la República, Montevideo 12900, Uruguay; 4Instituto de Investigación Biomédica de Bellvitge, 08907 Hospitalet del Llobregat, Spain; rauldominguez86@hotmail.com (R.D.); carlamarcocazcarra@gmail.com (C.M.); 30058mpp@gmail.com (M.P.); 5Institute of Neurosciences, Universitat Autònoma de Barcelona, 08193 Barcelona, Spain; 6Department Cell Biology, Physiology and Immunology, Universitat Autònoma de Barcelona, 08193 Barcelona, Spain; 7Centro de Investigación Biomédica en Red Sobre Enfermedades Neurodegenerativas (CIBERNED), Instituto de Salud Carlos III, 28031 Madrid, Spain

**Keywords:** amyotrophic lateral sclerosis, lipidomics, oxylipin, specialized pro-resolving mediators, mass spectrometry

## Abstract

Oxylipins play a critical role in regulating the onset and resolution phase of inflammation. Despite inflammation is a pathological hallmark in amyotrophic lateral sclerosis (ALS), the plasma oxylipin profile of ALS patients has not been assessed yet. Herein, we develop an oxylipin profile-targeted analysis of plasma from 74 ALS patients and controls. We found a significant decrease in linoleic acid-derived oxylipins in ALS patients, including 9-hydroxy-octadecadienoic acid (9-HODE) and 13-HODE. These derivatives have been reported as important regulators of inflammation on different cell systems. In addition, some 5-lipoxygenase metabolites, such as 5-hydroxy- eicosatetraenoic acid also showed a significant decrease in ALS plasma samples. Isoprostanes of the F2α family were detected only in ALS patients but not in control samples, while the hydroxylated metabolite 11-HETE significantly decreased. Despite our effort to analyze specialized pro-resolving mediators, they were not detected in plasma samples. However, we found the levels of 14-hydroxy-docosahexaenoic acid, a marker pathway of the Maresin biosynthesis, were also reduced in ALS patients, suggesting a defective activation in the resolution programs of inflammation in ALS. We further analyze oxylipin concentration levels in plasma from ALS patients to detect correlations between these metabolites and some clinical parameters. Interestingly, we found that plasmatic levels of 13-HODE and 9-HODE positively correlate with disease duration, expressed as days since onset. In summary, we developed a method to analyze “(oxy)lipidomics” in ALS human plasma and found new profiles of metabolites and novel lipid derivatives with unknown biological activities as potential footprints of disease onset.

## 1. Introduction

Amyotrophic lateral sclerosis (ALS) is a heterogeneous neurodegenerative disease characterized by the degeneration of both upper and lower motor neurons, leading to motor and extra-motor symptoms [1]. In most patients, the cause of ALS is unknown, although some individuals have a family antecedent, which is associated with mutations in genes that have a wide range of roles. Some of the implicated genes are incompletely penetrant in familial ALS, and with rare exceptions, genotype does not necessarily predict phenotype [1]. As is the case with other neurodegenerative diseases, it is now recognized that ALS is a heterogeneous condition associated with more than one pathogenic mechanism and with different clinical characteristics [2]. Degeneration of motor neurons is accompanied by neuroinflammatory processes, with the proliferation of astroglia, microglia, and oligodendroglia [3]. Neuroinflammation has been observed by imaging studies in patients with ALS, human post-mortem samples, and rodent models of ALS [4]. Moreover, biomarkers of neuroinflammation such as MCP-1, CHIT1, and YKL-40, are elevated in patients with ALS and have been shown to correlate with disease severity and predict disease progression [5,6]. Growing evidence suggests that, in addition to microglia, several other innate immune cell subsets, including macrophages, monocytes, dendritic and natural killer cells, mastocytes, and neutrophils are implicated in the ALS pathogenesis [7]. Indeed, the rate of disease progression in ALS has been suggested to be related to the degree of systemic monocyte/macrophage activation [8].

A large body of evidence shows that systemic inflammatory response can alter central inflammation in degenerative processes. Pro-inflammatory responses are present in the periphery in addition to neuroinflammatory disease within the central nervous system (CNS) [9]. Levels of circulating cytokines are abnormal in ALS patients, with higher levels of TNF-α, IL-1 β, IL-2, IL-8, IL-12p70, IL-4, IL-5, IL-10, and IL-13, and lower levels of INF-γ when compared to healthy controls [10]. IL-6 was also strongly associated with C-reactive protein levels and was found to increase the expression toward end-stage disease in a longitudinal analysis [10]. 

As ALS is a clinical disease with a heterogeneous phenotypic manifestation and clinical course, diagnostic and prognostic biomarkers are needed for stratification [1]. Neurofilament light polypeptide and phosphorylated neurofilament heavy polypeptide in cerebrospinal fluid (CSF) or blood is probably the most promising biofluid marker of ALS. Its levels correlate with the rate of disease progression and are most strongly associated with the involvement of the upper motor neuron system [1,11]. However, integration into standard clinical practice represents major difficulties [1]. Several reports investigated other potential biomarkers. A large primary care cohort study identified novel metabolic markers including alterations in carbohydrate, lipid, and apolipoprotein profiles, that were associated with an increased risk of ALS in later life [11]. Regarding inflammation, a recent meta-analysis involving 25 studies on peripheral cytokines levels in ALS confirmed that TNF-α, IL-1 β, IL-6, IL-8, and VEGF measured in blood were significantly elevated in ALS cases compared with controls [12].

Although inflammation is not ALS-specific, exploration of inflammation biomarkers may improve the knowledge of ALS pathogenesis. Regarding lipid mediators of inflammation, some evidence on alterations of PGE2 levels in ALS patients has been previously reported [13]. Eicosanoid analysis of ALS patient samples showed a significant increase in PGE2 levels in cerebrospinal fluid and also in postmortem spinal cord and cortex samples [14,15]. Recent reports in which the omics approaches have been applied found alterations in ALS patients’ lipid metabolism [16,17,18,19]. Henriques et al. [16] analyzed the HPLC profiles of fatty acids derived from total lipids in serum and clotted blood cells from ALS patients and controls and reported as a major conclusion that blood cell 16:1/16:0 ratio and palmitoleate levels correlate with disease progression. Blasco et al. [17] performed a lipidomics analysis of CSF from ALS patients searching for diagnostic and predictive footprints. They found predictive models for the variation of ALSFRS-r score from the lipidome baseline with 71% accuracy. Besides, significant prediction of clinical evolution correlated with sphingomyelins and triglycerides with long-chain fatty acids. A recent article by Area-Gomez et al. [18] reports a lipidomic analysis (complex lipids) of plasma from patients with ALS and primary lateral sclerosis (PLS, a more benign form of motor neuron disorders that only affects upper motor neurons). They found that common aspects of these pathologies suggest that PLS and ALS behave as a part of a continuum of motor neuron diseases from the lipidomics point of view. Another study from O’Reilly et al. [19] evaluated the association between pre-diagnostic plasma polyunsaturated fatty acids (PUFA) levels and ALS. They found that the majority of individual PUFAs were not associated with ALS, although in men linoleic acid (LA) was inversely correlated with the risk of ALS, and docosahexaenoic acid (DHA) was positively correlated to the risk of ALS. In women, they found that arachidonic acid (AA) was positively related to the risk. However, these findings warrant confirmation in further studies. Recently Fernandez-Eulate et al. [20] have reported a lipidomic analysis in sera from ALS patients and healthy volunteers, in which hydroxyl- derivatives from LA were measured. They claim one of the most comprehensive lipid profiling of ALS, which did not reveal any serum lipid signature discriminating patients with ALS from controls. To our knowledge, there is no lipidomic analysis containing a broad spectrum of oxylipins, applied to human samples from ALS patients. Thus, it is currently unknown whether the levels of oxylipins in the circulating plasma could reflect the ALS inflammatory component of the disease. In the present study, we performed a targeted liquid chromatography-mass spectrometry (HPLC-MS/MS) analysis to detect and quantify a full panel of oxylipins in plasma samples from 78 ALS and age-matched control healthy individuals.

## 2. Materials and Methods

### 2.1. Materials

All lipids were purchased from Cayman Chemicals, and specific working mixes were prepared by dilution with HPLC grade methanol. Internal standard mix (IS mix) contains 500 pg/µL of each of the following deuterated standards: (d4) 6k PGF1α, (d4) TxB2, (d4) PGF2α, (d4) PGE2, (d4) PGD2, (d8) 15-HETE, (d8) 12-HETE, (d8) 5-HETE, (d4) 8-isoPGF2α, (d11) 5-isoPGF2α, (d5) RvD1, (d5) MaR1, (d4) RvE1, (d4) 13-HODE and (d4) 9-HODE. 

### 2.2. Sample Preparation

Blood samples were collected in Barcelona according to The Research Protocol for Biosamples approved by the Ethics Committee of the Bellvitge University Hospital. Informed consent was obtained from all participants. Plasma samples were obtained between 3 and 66 months after the beginning of symptoms, and ALS was diagnosed according to the El Escorial criteria for ALS. The ALS-functional rate was calculated and validated by two independent neurologists. The progression rate was calculated (at baseline or last visit) as 48 minus the ALS Functional Rating Scale-Revised score, divided by the disease duration from onset of symptoms. 

Plasma was separated by centrifugation, immediately frozen and kept at −80 °C. Oxylipin analysis was performed by the MS platform of CEINBIO in Montevideo, Uruguay. Therefore, plasma samples were shipped in dry ice to CEINBIO and stored at −80 °C until analysis. 

Sample processing was performed as described previously [21]. Briefly, plasma samples are allowed to thaw on ice, and 500 µL aliquots are spiked with 2.5 µL of IS mix and diluted in an equal volume of 20% MeOH prepared in acidified water (HCl, pH 3). Tubes are gently vortexed and centrifuged for 30 min at 14,000× *g*, 4 °C. Clear supernatant is directly transferred to preconditioned SPE cartridges (Strata™-X, Polymeric sorbent 33 µ, 60 mg, 3 mL; Phenomenex). After the vacuum-assisted flow of the samples is achieved, cartridges are washed with 10% MeOH solution in acidified water (HCl, pH 3) and submitted to high vacuum (~11 In Hg) for no more than 5 min. Finally, elution of the oxylipins is performed with 1 mL MeOH, and tubes are transferred to SpeedVac to allow evaporation to dryness. Before HPLC-MS/MS analysis, samples are reconstituted in 50 µL of 50% MeOH in ultrapure water.

### 2.3. Oxylipin Analysis

Analysis and quantitation of oxylipins in plasma were performed by HPLC-MS/MS employing an Infinity 1260 HPLC system (Agilent), coupled to a hybrid triple quadrupole/linear ion trap mass spectrometer QTRAP4500 (ABSciex, Framingham, MA, USA). A complete list of analyzed oxylipins is shown in Appendix A.

Chromatographic separation was performed on a Luna 5µm C18(2) reversed-phase column (100 × 2.0 mm, 100 A, Phenomenex) using a security guard cartridge (C18, 4 × 3.0 mm, Phenomenex) at 30 °C. Solvent A consisted of 0.1% formic acid in ultrapure water and solvent B was 0.1% formic acid in acetonitrile. The flow rate was set to 300 µL/min and the gradient was as follows: 30% B from 0 to 0.1 min, linear increase to 95% B at 11 min, maintained at 95% B until 15 min, and back to 30% B in 0.1 min and maintained for 5 min.

For MS detection, the QTRAP4500 was operated in the negative ion mode and set in multiple periods of MRM experiments. All standards were first diluted in methanol for parameter optimization and directly infused into the mass spectrometer. Three main experiments were performed for each compound; first, a mass spectra employing the quadrupole 1 (Q1 experiment) and an enhanced resolution spectra using the ion trap (ER experiment), for m/z definition and then a fragmentation experiment (Product Ion experiment) to determine the fragmentation pattern and define possible MRM transitions. In all cases parameters were ramped to define the values that allow the best signal, finding declustering potential (DP) and collision energy (CE), which are the parameters with more influence on the signal quality (see all fragmentation spectra in Appendix A). Finally, flow injection analysis was performed using a mixture of ultrapure water and methanol with 0.1% formic acid as mobile phase, and parameters were adjusted to define the final HPLC-MS/MS method.

General parameters were set as follows: curtain gas (CUR): 50; collision gas (CAD): medium; ion spray voltage (IS): 4500 V; ion source temperature (TEM): 400 °C; nebulizer gas (GS1): 40; drying gas (GS2): 30; entrance potential (EP): −10; and collision exit potential (CXP): −13. Probe position (TIS) was set at 0.300 cm on the vertical axis and at 0.500 cm on the horizontal axis, electrode protrusion was 1–1.5 mm. Each oxylipin was monitored with at least three MRM transitions (Appendix A). Detection was carried out in four periods of MRM experiments: Period 1 from 0 to 8 min, Period 2 from 8 to 10.5 min, Period 3 from 10.5 to 13 min, and Period 4 from 13 min to end. This strategy allows increasing the time averaging for each data point (dwell time), thus decreasing the noise levels and increasing the signal-to-noise ratio. A full list of analytes contained in each period is shown in Appendix A along with the specific compound parameters. As a reference, Period 1 covers mostly triol fatty acids and COX-derived prostaglandins, Period 2 covers mostly diol fatty acids, Period 3 covers mostly monohydroxylated fatty acids, and Period 4 covers unmodified fatty acids. An exemplary chromatogram for all oxylipins is shown in Appendix A.

Instrument control and data acquisition were performed with Analyst 1.6.2 (ABSciex, Framingham, MA, USA) software. PeakView 2.2 and MultiQuant 2.1 (ABSciex, Framingham, MA, USA) software were employed for data visualization and quantitation purposes, respectively.

### 2.4. Quantitation

Quantitation was based on external calibration utilizing 15 deuterated internal standards covering all the chromatographic regions. Both natural and isotope-labeled standards for most oxylipins were assessed to construct calibration curves ranging from LOD to 500–2500 pg, plotting the peak area ratio (analyte/IS) against the mass ratio (pg, analyte/IS) (see Appendix A for calibration curves details). Those oxylipins for which standards were not available were quantified employing calibration curves of the most similar molecule as calibrant (Appendix A).

### 2.5. General LOD and LOQ Determination

The limit of detection (LOD) and lower limit of quantification (LOQ) for all analyzed metabolites are shown in Appendix A. Along with calibration curves, blank samples (50–70 samples) containing IS mix were submitted to HPLC-MS/MS analysis, and standard deviation of area ratio was obtained for each analyte. LOD and LOQ values were calculated as the concentration in which signal corresponds to the blank plus 3 or 5 times the standard deviation of the blank, respectively. To translate into plasmatic LOD, dilution factor and % recovery were used. The dilution factor is 10, as oxylipins derived from 500 µL of plasma are eventually dried and resuspended in 50 µL prior analysis. Percentage recovery for each analyte was assumed as the %recovery of the internal standard employed as reference.

### 2.6. Specilized Pro-Resolving Lipid Mediators LOD and LOQ

To determine the limit of detection (LOD) and lower limit of quantification (LOQ), a standard mix containing all specialized pro-resolving mediators (SPM) standards available in our lab was prepared at a low concentration (lower values of calibration curves) and then diluted in half with methanol several times. Internal standard mix was added and then HPLC-MS/MS analysis was carried out. The Signal Finder algorithm (Multiquant 2.1, ABSciex) was employed for peak area integration of the most intense transition for each SPM. Then, LOD and LOQ were defined as the mass (pg) at which the signal-to-ratio (S/N) was equal or greater than 3 or 5, respectively (see Appendix A for complete data). 

### 2.7. Statistical Analysis

For individual oxylipin analysis, a non-parametric Mann Whitney´s test was used for single comparisons between Control and ALS groups, performed in GraphPad Prism 8 (GraphPad Software, San Diego, CA, USA). Tukey plot of plasmatic concentration of selected oxylipins and correlation plots were performed with the same software.

Other statistical analyses were performed in the statistical package R [22]. In an attempt to identify combinations of oxylipins concentrations that could discriminate between groups of patients we performed several oxylipins multivariate exploratory analyses, including hierarchical and non-hierarchical (k-means) clustering, metric and non-metric multidimensional scaling, discriminant analysis, and orthogonal partial least squares discriminant analysis (OPLS-DA).

Non-parametric analysis of variance (Kruskal–Wallis test) was used to detect significant differences in continuous variables according to patient groups. Correlations between continuous variables were assessed using the Pearson correlation test after logarithmic scaling (in particular for oxylipins) with posterior Benjamini-Hochberg correction for multiple testing.

## 3. Results

### 3.1. Oxylipin Analysis

Plasma samples of ALS patients, as well as healthy volunteers, were obtained from the Biobank HUB-ICO-IDIBELL (PT17/0015/0024) and the Motor Neuron Unit of Bellvitge University Hospital. Control samples were obtained from patients without any type of neurological disorder. The demographic and clinical characteristics of donors are shown in Table 1. Frozen plasma samples were analyzed by HPLC-MS/MS to detect and quantify free oxylipins. Analyzed oxylipins include LA, AA, eicosapentaenoic acid (EPA), and DHA metabolites, derived from enzymatic and non-enzymatic pathways. A full list of analyzed metabolites is shown in Appendix A. 

### 3.2. Individual Oxylipin Analysis

Plasmatic concentration levels for all detected oxylipins are shown in Table 2 and a comparison analysis between the Control group and ALS patients was performed (Mann–Whitney test). As shown in Table 2 and Figure 1A,B, LA-derived metabolites show a significant decrease in ALS patients compared to control samples. Particularly, lipoxygenase derivatives show a decrease by 50%, while CYP450 derivatives show an even more considerable reduction (~3–9 fold). Regarding the AA pathway, it should first be noted that there is no significant difference in the plasmatic concentration of the precursor (Figure 1B). However, some 5-lipoxygenase metabolites (Figure 1A,B) such as 5-HETE and 5-oxoETE show a significant decrease in ALS patients, while for others such as LTB4 the reduction does not reach statistical significance (*p*-value = 0.07). Similarly, 12-hydroxyeicosatetraenoic acid (12-HETE) also indicates a decrease in ALS patients although the significance is not achieved. Regarding the non-enzymatic pathways, isoprostanes of the F2α family were detected but only in 17 ALS patients. Interestingly, the hydroxylated metabolite, 11-HETE was detected in all samples, showing a significant decrease in ALS patients. As a result of this analysis, a schematic representation of major altered oxylipin pathways in plasma from ALS patients is shown in Figure 1C.

There are no significant differences in plasmatic-free fatty acids levels regarding omega 3 fatty acids, EPA and DHA. Detected omega 3 fatty acid derivatives consisted mainly in the monohydroxylated fatty acids and no significant differences were detected between ALS and Control groups for any of them (Table 2).

Despite our effort to detect SPM, these metabolites were not detected in ALS nor control plasma samples. Then, to evaluate the complete method for SPM detection in human plasma, we performed a supplementation experiment of SPM to fresh human plasma (which was previously analyzed and SPM levels were < LOD). An SPM mix containing LXA4, LXB4, RvE1, RvD1, RvD2, RvD3, RvD5, MaR1, and PD1 was spiked at different concentration levels (ranging from 0.1 to 10 nM, specific values shown in Appendix A). Then, both regular sample processing and HPLC-MS/MS analysis were performed. Detection of spiked SPM was verified for all compounds, even in the lower concentrations tested (0.1–0.5 nM, Appendix A), thus verifying the method’s ability to detect SPM in the low nM range. Despite SPM were undetected in plasma samples, we observed that the levels of 14-hydroxy-docosaexaenoic acid (14-HDoHE), one of the most abundant hydroxy-DHA derivatives detected in plasma and a key activation marker of the Maresin biosynthesis pathway, were significantly reduced in ALS patients, further suggesting ALS patients fail to switch on the production of lipids that are responsible to resolve inflammation, at least in the periphery.

### 3.3. Multivariate Analysis

Considering the detected oxylipins, we performed multivariate analysis (metric and non-metric multidimensional scaling, hierarchical clustering, k-means clustering, and discriminant analysis) in search of differences between the Control group and ALS patients. However, it was impossible to discriminate between patients and controls (data not shown). We also performed comparative tests between ALS subgroups, regarding phenotype (bulbar or spinal onset), progression rate, gender, and other clinical parameters (data not shown), where no significant differences were detected.

### 3.4. Correlation Analysis

We further analyzed oxylipins concentration levels detected in ALS patient samples, to assess the correlation between metabolites and clinical parameters. Logarithmic transformation of concentration values was performed and the Pearson correlation test evaluated the correlation between oxylipin concentration and phenotype, age, progression rate, and time since disease onset (elapsed time since onset to blood sample extraction) with Benjamini-Hochberg correction for multiple testing. As shown in Figure 2, plasmatic levels of LA-derived 13-HODE and 9-HODE, positively correlate with disease duration, expressed as days since onset (logarithmic transformation). No correlation was found between the concentration values of other oxylipins and the above-mentioned clinical parameters.

## 4. Discussion

A misbalance in pro- and anti-inflammatory and pro-resolving lipid mediators could play a critical role in neuroinflammation associated with ALS disease. Despite some evidence on alterations in AA-derived metabolites that have been reported, to our knowledge, no “(oxy)lipidomics” approach in ALS human samples has been performed. Herein, we developed an oxylipin profile analysis of plasma samples of ALS patients and healthy volunteers. 

Previous reports on eicosanoid analysis in ALS patient samples showed a significant increase in PGE2 plasma levels [13], and postmortem spinal cord and cortex tissue [14,15]. This was not observed in the samples analyzed here, emphasizing the relevance of disease onset conditions. Samples in previous work were from postmortem tissues or advanced ALS patients whereas our analysis included patients with only 1–3 years of disease onset. In addition, the reported determinations of PGE_2_ were obtained by immunoassays while here we performed a LC-MS analysis having greater specificity and sensitivity. 

Regarding SPMs, as demonstrated for other neurodegenerative diseases, e.g., Alzheimer´s disease [23] and multiple sclerosis [24,25], lack of activation of resolution of inflammation programs is represented as a decrease in SPM levels which leads to sustained inflammation. We recently showed that the production of the main lipid mediators that activate resolution programs of inflammation is deficient in serum and active brain lesions of multiple sclerosis patients, as well as in the spinal cord of experimental autoimmune encephalomyelitis mice [25]. Furthermore, these observations were accompanied by an increase in AA-derived pro-inflammatory lipid mediators, such as LTB4, PGE2, PGF2 and TXB2, emphasizing an imbalance between pro-inflammatory and pro-resolving lipid mediators [25]. Thus, we hypothesized that a similar scenario could be responsible for the persistent inflammatory component in ALS. We developed an HPLC-MS/MS method [21,25] based on previous reports [26,27,28,29,30] to study plasma oxylipin profiles in ALS patients. As described in the Material and Methods section, this method comprises sample processing by solid-phase extraction and reversed-phase liquid chromatography before mass spectrometry detection. However, no SPMs were detected either in Control or ALS groups despite our effort. We then analyzed plasma samples spiked with a mixture of SPM and detection was performed for all SPMs, even in pg/mL concentration values (Appendix A). Values of SPMs concentration in plasma from healthy individuals range from 1 to 2000 pg/mL in some reports [31,32] while others have reported levels <LOD when analyzing samples from healthy individuals [27]. However, several difficulties are present when analyzing SPM (see comprehensive review [33] for examples of SPM quantitation issues). Furthermore, detection in healthy individuals has been questioned in the literature [34]. Despite the continuous effort from individual laboratories and international scientific societies, there is still a lack of consensus on sample preparation procedures and/or routine performance standard that allows high confidence in absolute values for oxylipin concentration [28,35]. Then a comparison between inter-laboratories is still challenging. Nonetheless, intra-laboratory comparison of measurements, i.e., between our ALS and Control groups, is reliable regarding method validation. Even considering the above-mentioned observations, SPM concentration in plasma should be expected to augment upon the inflammation challenges. However, plasma samples of ALS patients behave similar to the Control group, being non-possible to detect SPMs. The lack of detection of SPM could indicate a defective resolution program of systemic inflammation in ALS. Although not fully demonstrated, the hypothesis that a resolution program should be activated in the inflammatory state of ALS requires further investigation. 

A significant decrease in LA-derived oxylipins in ALS patients was revealed in this study. The biosynthetic pathway of the monohydroxy derivatives comprises 12-lipoxygenase action, which oxidizes LA at C9, or 15-lipoxygenase-1 action, which oxidizes at C13, resulting in the formation of 9-hydroperoxy-octadecadienoic acid (9-HpODE) and 13-HpODE, respectively. These peroxy- derivatives are unstable and are reduced to the hydroxyl form by glutathione peroxidases forming a 9-HODE and 13-HODE [36]. The biological role of these metabolites is not fully understood; however, among other LA derivatives, HODEs have been reported as predominant regulators of inflammation on different cell systems. Specifically, HODEs are involved in the regulation of cell adhesion to the vascular wall in an inflammatory context, showing beneficial effects downregulating cell adhesion. It has been proposed that 13-HODE plays a role in down-regulating adhesion molecule expression by endothelial cells (as opposed to pro-inflammatory cytokines, e.g., IL-1α). Moreover, HODEs have also been shown to decrease PGE2 production by endothelial cells [36]. Our results are in line with those reported by Fernandez-Eulate et al. [20], who report a decrease in both HODE isomers in ALS patients compared to Control. They attribute this difference to the confounding effect of lipid-lowering statins, however, in our ALS group we are not aware of statin treatment in ALS patients. Thus, further investigation is needed to confirm the alteration in HODEs plasmatic levels in ALS patients and its role in peripheral inflammation. It is worth mentioning that contrary to the results obtained for ALS patients, HODEs are increased in the blood of AD patients and were suggested by the authors as a potential biomarker for the diagnosis of AD [37]. 

LA might also be metabolized by CYP pathways and converted to epoxides 9,10-EpOME and 12,13-EpOME, also known as leukotoxin and isoleukotoxin, respectively. These epoxides are then metabolized principally by soluble epoxide hydrolase to their corresponding diols, 9,10-DiHOME and 12,13-DiHOME, named leukotoxin diol and isoleukotoxin diol, respectively. Several biological functions have been attributed to these CYP-derivatives. Since its first description as toxins produced by leukocytes (which gave its name) many functions have been ascribed to EpOMEs and DiHOMEs. Regarding inflammation and immune response, it has been shown that neutrophils produce DiHOMEs and these metabolites act as chemoattractants of other neutrophils at relatively lower doses (~10 nM). At the micromolar range, it has been shown that it inhibits neutrophil respiratory burst in cell cultures, although it does not share the exact mechanism of LXA4. These observations suggest that biosynthesis of DiHOMEs may serve as a type of negative feedback that limits the inflammation; however, these metabolites are not considered as pro-resolving mediators. Recent research revealed an additional for DiHOMEs. It was found that both DiHOME isomers are increased after cycling exercise and are subsequently reduced back to baseline after exercise is over [38]. Moreover, HODEs were positively correlated with 12,13-DiHOME postexercise levels, along with isoprostanes, which led to propose them as biomarkers of oxidative stress [38]. Additionally, it was found that routinely active subjects had significantly higher pre-exercise 12,13-DiHOME levels compared to subjects who did not regularly exercise [39,40]. Then, our results showing a decrease in DiHOMEs in ALS patients might reveal either a diminished activation of negative feedback inflammatory regulation or simply an indicator of the diminished physical activity in ALS patients. Our results show that ALS patients have diminished HODE and DiHOME plasmatic levels compared to control samples. However, we found that HODEs positively correlated with time since the onset of disease, which requires further investigation.

In conclusion, this work shows for the first time a HPLC-MS/MS profile analysis of oxylipins in plasma of patients with ALS and healthy volunteers. Despite current advances in the characterization of peripheral inflammation in ALS, we did not detect a discriminant profile of lipid mediators of inflammation between patients and controls. However, in individual analysis, we have identified significant differences for some oxylipins i.e., LA-derived oxylipins and 5-lipoxygenase metabolites of AA. A decrease in lipoxygenase and CYP450-derived metabolites from LA is promising as a potential biomarker of ALS. However, further investigation is still required to elucidate this decline’s biological role and significance.

## Figures and Tables

**Figure 1 biomedicines-10-00674-f001:**
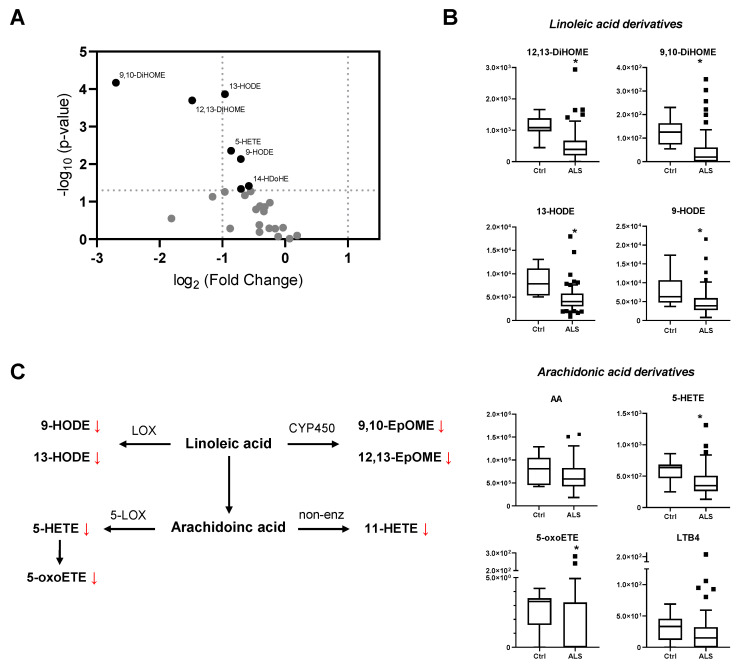
Individual oxylipin analysis. (**A**) Volcano plot representation indicating the −log_10_ (*p*-value) and the log_2_ (fold change) of the oxylipin concentration in plasma for ALS vs. Control group shown in Table 1. Dots in black represent *p* < 0.05. (**B**) Tukey plot for plasma concentration (pg/mL) of selected oxylipins in Control (Ctrl) and ALS patients (ALS). Upper panels, LA-derived metabolites were detected and a significant decrease is shown for ALS plasma samples. Lower panels, arachidonic acid, and 5-lipoxygenase-derived metabolites plasma concentration. (**C**) Scheme representing major altered oxylipin pathways in plasma of ALS patients. Significantly altered metabolites are derived from LOX and CYP450 pathways. * *p* < 0.05, Mann–Whitney test.

**Figure 2 biomedicines-10-00674-f002:**
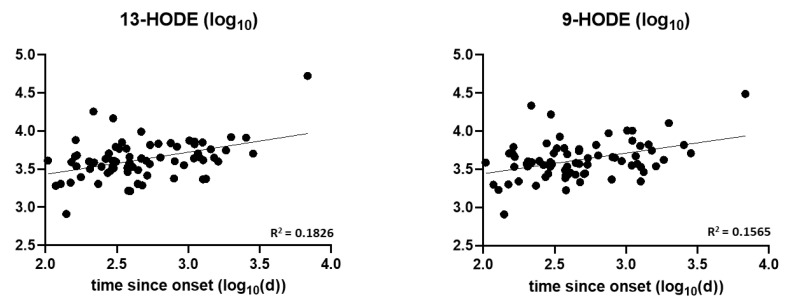
Correlation plot between LA metabolites and disease duration in ALS patients. Pearson correlation tests and *p*-value adjustment (Benjamini-Hochberg) were performed for all detected oxylipins and clinical parameters. A positive correlation was found for 13-HODE and 9-HODE with time since onset (logarithmic transformation).

**Table 1 biomedicines-10-00674-t001:** Demographic and clinical characteristics of ALS patients and control volunteers.

	ALS (*n* = 78)	Control (*n* = 9)
**Gender**		
Female	31	7
Male	47	2
**BMI** ^a^	25.2 (+/− 3.8)	23.2 (+/− 1.8) ^b^
**Site at onset**		
Spinal	42	-
Bulbar	28	-
Other	8	-
**Age**		
at onset (years) ^a^	62.2 (+/− 11.8)	-
sample collection (years) ^a^	63.8 (+/− 11.1)	47.3 (+/− 15.8)
**Time since onset (months)** ^a^	23.1 (+/− 30.9)	-
**ALSFRS-r slope at last visit (*n* = 65)**	1.35 (+/− 1.61)	-
SP	28	-
NP	17	-
FP	24	-

^a^ Values are given as mean ± standard deviation. ^b^ Data obtained for *n* = 5 control volunteers.

**Table 2 biomedicines-10-00674-t002:** Plasma oxylipin concentrations in Control and ALS samples assessed by HPLC-MS/MS.

Metabolites	Control (*n* = 9)		ALS (*n* = 74)		*p*-Value ^b^
	Median (pg/mL)	Percentile ^a^	Median (pg/mL)	Percentile	
**LA (18:2n-6) metabolome**					
*12,13-DiHOME*	*1084*	*(969.3; 1381)*	*389*	*(201; 666)*	*0.0002*
*9,10-DiHOME*	*126*	*(73.1; 163)*	*19.4*	*(1.20; 60.0)*	*<0.0001*
*13-HODE*	*7865*	*(5375; 11,132)*	*4049*	*(3044; 5773)*	*0.0001*
*9-HODE*	*6332*	*(4769; 10,706)*	*3881*	*(2775; 5951)*	*0.0073*
**AA (20:4n-6) metabolome**					
AA	811,395	(457,556; 1,046,666)	588,400	(429,801; 825,059)	0.1601
TxB2	440	(46.9; 614)	125	(46.6; 412)	0.2800
PGE2	0	(0; 7.73)	0	(0; 0)	0.6111
PGD2	0	(0; 0)	0	(0; 0)	0.9029
isoPGF2	0	(0; 0)	0	(0; 443)	0.1935
*5-HETE*	*636*	*(469; 685)*	*350*	*(261; 505)*	*0.0044*
8-HETE	124	(83.0; 137)	84.7	(58.7; 117)	0.0531
*11-HETE*	*191*	*(109; 224)*	*117*	*(87.9; 167)*	*0.0459*
12-HETE	3144	(2673; 4431)	1616	(771; 4125)	0.0550
15-HETE	247	(181; 271)	187	(139; 247)	0.1312
*5-oxoETE*	*3.29*	*(1.61; 3.52)*	*0*	*(0; 3.22)*	*0.0089*
12-oxoETE	36.4	(0; 59.8)	0	(0; 42.3)	0.4171
15-oxoETE	67.9	(0; 119)	37.0	(27.7; 63.5)	0.5207
14,15-DiHETrE	379	(321; 446)	320	(269; 383)	0.1064
LTB4	33.2	(11.9; 45.5)	14.9	(0; 32.1)	0.0741
**EPA (20:5n-3) metabolome**					
EPA	198,372	(121,928; 361,172)	208323	(146,448; 292,417)	0.9713
12-HEPE	117	(90.1; 169)	92.7	(0; 157)	0.1816
**DHA (22:6n-3) metabolome**					
DHA	329,956	(189,149; 560,821)	376709	(261,062; 529,711)	0.8010
4-HDoHE	260	(158; 297)	196	(147; 283)	0.4142
7-HDoHE	75.6	(0; 111)	69.9	(0; 102)	0.8521
10-HDoHE	27.2	(20.0; 33.9)	24.3	(18.8; 31.1)	0.5217
11-HDoHE	30.0	(19.9; 39.2)	23.9	(16.9; 30.1)	0.1384
13-HDoHE	87.4	(45.3; 125)	56.1	(33.9; 81.3)	0.0675
*14-HDoHE*	*117*	*(86.5; 222)*	*78.5*	*(41.1; 127)*	*0.0381*
16-HDoHE	66.9	(37.5; 102)	56.1	(43.4; 84.9)	0.5114
20-HDoHE	31.2	(10.7; 44.0)	23.5	(16.1; 35.9)	0.6488
19,20-DiHDPA	160	(149; 201)	157	(136; 193)	0.4929

^a^ 25% percentile and 75% percentile are shown; ^b^ derived from the Mann–Whitney test. Metabolites in italics are significantly different between groups according to the Mann–Whitney test.

## Data Availability

The data presented in this study are available on request from the corresponding author.

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
