# Peer review of "HPLC-MS/MS Oxylipin Analysis of Plasma from Amyotrophic Lateral Sclerosis Patients"

_biomedicines, 2022, doi:10.3390/biomedicines10030674_

Round 1

Reviewer 1 Report

In this manuscript by Mauricio et al, authors analyzed oxylipin profiles in plasma of ALS patients and found that levels of 13-HODE and 9-HODE positively corelate  to disease duration and concluded lipid metabolite profile as potential footprint of disease onset, decrease in lipoxygenase and CYP450 derived metabolites as a promising biomarker for ALS.

comments:

1) Authors need to address the significant age difference between control and ALS group of samples

2) BMI information of control group is important as it effects the lipidomics of individual

3) Figure 1 data can be better represented by using scatter plot incorporating both ALS and control samples. Please refer the journal below

"A comprahensive serum lipidome profiling of amyotrophic lateral sclerosis".

4) A schematic represenation of differences in lipidomics profile between control and ALS samples is needed 

Reviewer 2 Report

Despite current advances implying peripheral inflammation in ALS, this paper fails to detect a discriminant profile of lipid mediators of inflammation between patients and controls. However, in individual analysis, they identified significant differences for linoleic acid-derived oxylipins and 5-lipoxygenase metabolites of arachidonic acid. A decrease in lipoxygenase and CYP450 derived metabolites from LA is promising as a potential biomarker of ALS.

Round 2

Reviewer 1 Report

The manuscript is overall improved after revision